# New Insights into the Medieval Hispano-Muslim Panel Painting: The *Alfarje* Found in a Balearic *Casal* (Spain)

**DOI:** 10.3390/molecules28031235

**Published:** 2023-01-27

**Authors:** Carla Álvarez-Romero, Ana García-Bueno, Teresa López-Martínez, Rafael Turatti-Guerrero, Noemí Montoya, María Teresa Doménech-Carbó

**Affiliations:** 1Painting Department, University of Granada, Avenida de Andalucía 27, 18014 Granada, Spain; 2Institut Universitari de Restauració del Patrimoni, Universitat Politècnica de València, Camí de Vera 14, 46022 València, Spain; 3Trívium C. B, Calle Ceniceros 25, 18010 Granada, Spain; 4Health Sciences School, Universidad Internacional de la Rioja, UNIR Avenida de la Paz 137, 26006 Logroño, Spain

**Keywords:** Hispano-Muslim art, panel painting, indigo, panel painting alteration, metal soap, calcium oxalate, FIB-FESEM-EDX, AFM-NI, FESEM-EDX, FTIR spectroscopy

## Abstract

Hispano-Muslim culture flourished during the Middle Ages in the Iberian Peninsula and Balearic Islands. During the restoration of a Balearic nobiliary building (*casal*), several panels with polychrome decoration on the back side were found. They were part of an old Muslim wooden ceiling (*alfarje*). A multi-technique strategy including optical microscopy, infrared and μRaman spectroscopies, field emission scanning electron microscopy-X-ray microanalysis (FESEM-EDX), focused ion beam (FIB-FESEM-EDX), atomic force microscopy nanoindentation (AFM-NI), and gas chromatography-mass spectrometry (GC-MS) has been applied in the analysis of these panel paintings and has provided morphological and compositional data that have led to the identification of the materials and artistic technique as well as the alteration mechanisms due to the natural aging and the adverse conditions of conservation. As a novelty, this study has confirmed the use of indigo as a blue pigment, an unusual material in Hispano-Muslim panel painting. Apart from the notable change in the visual appearance observed in the paintings, the study has also confirmed a change in the mechanical resistance in the paint layers. These changes have been induced by the combination of the chemical and microbiological alteration mechanisms identified.

## 1. Introduction

The Balearic Islands have always been in the crosshairs of the cultures present in the western Mediterranean. In the Middle Ages, during the Muslim domination which lasted more than four centuries, this geostrategic area witnessed frequent power struggles. Isam al-Jawlani overrated the islands in the 10th century and started Islamization. After this, several crises took place. The annexation to the Caliphate of Córdoba [1,2,3]; the incorporation in 1014 into the Taifa of Denia [1,2,3,4]; several Crusader expeditions (1107–1108; 1113–1114) [1,2,3,4]; the rise of Almoravids (1114) [1,2,3,4]; and the Almohad conquest of Mayurqa (1203) [1,2,3,4,5]. The Islamic domination finished with the conquest of the island by King Jaime I of the Crown of Aragon in 1229 [1,2,3,4,5]. In this period, the city of Palma de Mallorca experienced significant growth and urbanization, as did the island’s rural areas. 

During the restoration of a historical building in Palma de Mallorca, several panels with the back side polychromed were found as part of an old wooden ceiling (*alfarje*) (Figure 1). The habit of reusing materials during the first reconstruction of the city in the further Christian period and the subsequent reforms prevented the destruction of those wooden panel paintings. The building where the panels were found is located in the old quarter of the city of Palma de Mallorca, on Caputxins 9 street. Its location is a heritage setting, surrounded by houses of Gothic origin, between the Capuchinas convent and the church of San Jaime. It is a Majorcan typical mansion (*casal*), with the classic structure of a noble house from the Modern Age. Its construction, which would occupy a large area in the urban complex of the old Madina Mayurqa, began in medieval times, probably in the 12th century, as an Islamic fortified house belonging to a nobiliary family. The area was crucial as a defensive place, as it dominated the lower part of the city, maintaining that distinction throughout history. In addition to the defensive function of the fortification, the building was attached to one of the five mosques that the city had, which encouraged King Alfonso III of Aragon to identify them as a military objective and, at the end of the 13th century, ordered it to be burned and subsequently reconstructed. The building was reformed in the Baroque style between the 17th and 18th centuries. Currently, the building is registered in the Local Catalogue of Protected Built and Natural Heritage.

After the reconquest of the Balearic Islands by King Jaime I, Christian, Jewish, and Muslim people lived together, which led to a fruitful exchange of knowledge in many areas, including architectural and artistic knowledge. Therefore, while Romanesque and Gothic styles were being developed in continental Europe, singular Hispano-Muslim and its successor, *Mudéjar* style (*Mudejar* people were Muslims living in Christian territory after the reconquest), were developed in the Iberian Peninsula Kingdoms. Hispano-Muslim styles, particularly *Mudéjar* style, were characterized by the mixture of Romanesque and Gothic architectural elements employed by Christians and ornamental motifs and materials used by Hispano-Muslim and Jewish artisans [6,7,8].

The wooden panel paintings in the Palma de Mallorca *casal* are an excellent example of *alfarjes*, the medieval wooden ceilings made in Hispano-Muslim style. Two wooden ceiling styles were used in the medieval period: the *alfarjes* and the *armaduras*. The *alfarjes* is the most straightforward system for decorating ceilings and consists of wood panels fitted parallel to a flat ceiling. This word comes from the Arabic verb *faraŝa*, which means upholster or hang [9]. The *armaduras* is a gabled ceiling. These covering systems have been used throughout the Mediterranean area since Roman times. However, they come from different geographical contexts; the *armaduras* from the Visigothic tradition, from the Western Roman Empire, and the *alfarje* from the Muslim tradition, from the Eastern Roman Empire [10]. In addition, the use of colors for decorating paneled ceilings is another characteristic of Hispano-Muslim and *Mudéjar* artistic styles. Polychromed motifs decorated all types of architectural elements: pavements, baseboards, vaults, arches, latticework, or walls [11].

The conservation of this type of cultural heritage is fundamental due to its scarcity and importance. On the one hand, it must consider the wooden structure. Due to their organic nature and hygroscopic properties, wooden structures present great vulnerability [12,13,14]. On the other hand, paintings also tend to present a series of pathologies and alterations that mean that, over time, their physical, mechanical, or visual appearance has changed, and they do not appear as they were initially conceived [15,16,17,18].

The use of analytical instruments applied to cultural heritage allows for obtaining necessary information about the works of art, including the nature of supports, pigments, binding media, varnishes, and other protective layers used by the artists. These studies also permit increasing knowledge about their state of preservation by showing the alterations the works have undergone over the years. When all this information is combined, a multidisciplinary interpretation of the results is possible through evaluating materials and techniques. Attribution studies, detection of forgeries, and selection of optimal conservation strategies are also goals of analytical studies on polychromed artworks.

Analytical techniques have been used in the past on Hispano-Muslim and *Mudéjar* monuments and artworks. The plasterwork of Dar al-Manjarra al Qubra (Royal Room of Santo Domingo, Granada, a fundamental monument for the definition of Nasrid art, antecedent of the Alhambra) was examined to identify the pigments, binding media, and the mortars through optical microscopy, scanning electron microscopy-X-ray microanalysis (SEM-EDX), gas chromatography-mass spectrometry (GC-MS), high-pressure liquid chromatography (HPLC), and X-ray diffraction (XRD) [19]. Several studies have been carried out in different parts of the Alhambra (Granada), the grandest and finest example of Islamic art and architecture from the Middle Ages still standing in the western world. Studies on the polychrome carpentry in the hall of the Mexuar Palace were carried out through optical microscopy, SEM-EDX, and GC-MS [20]. The study of the binding media used in the plasterworks from the hall of the king and the hall of the beads was performed by GC-MS [21]. The pigments used in the Lions Courtyard and the Hall of Ambassadors were studied through a multi-technique strategy that combines optical microscopy with SEM-EDX, Fourier transform infrared (FTIR), μFTIR, Raman and μRaman spectroscopies, and GC-MS [22,23,24,25]. *Mudéjar* monuments, such as the Casa de Pilatos palace (Seville), have been analyzed by portable X-ray fluorescence (XRF), SEM-EDX, optical microscopy, μRaman, and μFTIR [26]. In the study of the plasterworks and the façade in the Palace of King Pedro I and different courtyards of the Royal Alcázar of Sevilla [27,28,29,30,31,32] optical microscopy, SEM-EDX, XRD, GC-MS, Raman, and FTIR spectroscopy were used. 

In this context, a multi-technique strategy has been applied, which includes microscopy, spectroscopy, and chromatography techniques, intending to identify materials and artistic techniques used by the craftsman of the Palma de Mallorca *alfarje* as well as the alteration processes that have occurred due to natural aging and adverse conservation conditions. The results obtained can help to make proper decisions on the conservation treatment of this valuable cultural good.

## 2. Results

The description of the panel paintings and the criteria followed for establishing the sampling strategy are presented in Section 4.1 and Section 4.2.

### 2.1. Paint Strata Distribution

Table 1 summarizes the strata distribution, assignment of type of layers, thickness, and compounds identified from the experimental results obtained with the instrumental techniques employed. Most representative examples of strata distribution are shown in Figure 2. A complete set of images of the analyzed samples, obtained with different optical configurations in the microscopes, are supplied in Appendix A, which also includes a description of the morphological and optical properties of the pigments identified in each layer.

### 2.2. Pigments and Other Inorganic Materials Used in the Polychromies

#### 2.2.1. Optical Microscopy

As shown in Table 1, five pigments that correspond to white, yellow, red, deep blue, and blue-greenish colors compose the artist’s color palette. Although the chemical composition of the pigments has been confirmed with field emission scanning electron microscopy- X-ray microanalysis (FESEM-EDX), FTIR, and μRaman spectroscopy, direct examination in the optical microscope has provided a first recognition and characterization of their microscopic morphology. Lead white (2PbCO_3_·Pb(OH)_2_) grains of pigment can be observed in several cross-sections shown in Figure 2b,e–g. This pigment exhibits coarse grains that consist of large polycrystalline aggregates of fine particles. The aggregates look transparent and colorless and exhibit high birefringence. This morphology is characteristic of the Dutch or stack process used in the Middle Ages [15,33]. 

Figure 2g,h,j and Figure 3a show the image of natural azurite (Cu_3_(CO_3_)_2_(OH)_2_). The particles exhibit a blue hue ranging from deep blue to greenish-blue. The grains have angular shards, irregular shapes, and a size distribution ranging from coarse to fine due to the crushing process, which results in an uneven particle size distribution. Observed with polarized light, the particles exhibit pleochroism typical of azurite. In Figure 3b accessory minerals that usually accompany azurite, such as quartz, calcite, and clayey minerals, are also seen [15,16,17,18,19,20,21,22,23,24,25,26,27,28,29,30,31,32,33,34]. Natural cinnabar (HgS) is recognized by the characteristic intense red-orange color of its particles, which also exhibit high relief, prismatic habit, and angular shape due to the crushing process, as shown in Figure 2a–c and Figure 3c [15,35]. Indigo (C_16_H_10_N_2_O_2_) has been identified in several samples (Figure 2d–f). Small deep blue-greenish aggregates of fine particles of indigo are shown in Figure 3d. The grains appear opaque and rounded in shape [15,36]. In Figure 2d and Figure 3e can be seen particles of orpiment (As_2_S_3_). Optically this pigment is characterized by its straw-yellow color. Its particle size, which ranges from medium to fine, suggests that the pigment was artificially prepared by a dry sublimation process [15,37]. Finally, the inert material for preparing the ground can be observed in the cross-sections shown in Figure 2a,g–j. Figure 3f shows the morphology of gypsum (Ca(SO_4_)·2H_2_O) grains. The euhedral elongated-rhombic particles exhibit characteristic swallowtail-twinned or arrow-tail habits, with scattered particles rectangular in the shape of anhydrite [15]. The presence of the latter has been associated with a high burning of the raw material in the primitive kilns used in which the control of temperature was not perfect [38,39]. Micromorphology of lead white has been identified using focused ion beam-field emission scanning electron microscopy- X-ray microanalysis (FIB-FESEM-EDX) (vide infra).

#### 2.2.2. FESEM-EDX, µRaman and FTIR Spectroscopy

As previously mentioned, the chemical composition of the pigments was confirmed using FESEM-EDX, µRaman, and FTIR spectroscopy. Figure 4 shows several backscattered electron images of the cross-sections and the X-ray and µRaman spectra obtained in layers containing the pigments identified in the studied samples. More images and X-ray spectra are presented in Appendix A.

Use of cinnabar, orpiment, and tin and silver foil in samples (see Table 1) has been confirmed from the characteristic emission lines of S and Hg, S and As, Sn and Ag, respectively, that appear in the X-ray spectra. Averages of Hg/S and As/S molar ratios have been calculated for all the samples where cinnabar and orpiment were found from the quantitative measurements of the elemental composition performed with the FESEM-EDX. Average values of the Hg/S molar ratio range from 0.97 to 1.01 and properly match with the ideal formula of cinnabar (HgS). Similarly, the average values of the As/S molar ratio for orpiment were found to range between 0.69 and 0.71, in good agreement with the ideal formula of arsenic sulfide (As_2_S_3_) [40].

The presence of silver and tin foils was confirmed by the distinct emission lines of Ag and Sn that appeared in the X-ray spectra of samples J-4 and J-2. X-ray spectra of samples C-4, C-2, F-1, F-4, and D-1 exhibited characteristic emission lines of Pb that were assigned to lead carbonate, whereas the emission line of Cu was recognized in sample J-2. The morphological and optical properties observed in the microscopic examination, combined with the features observed in X-ray spectra, suggest the presence of lead white and azurite in these sample. Nevertheless, for the unambiguous confirmation of the use of these pigments in the *alfarje*, the samples were also analyzed by µRaman spectroscopy. Thus, bands at 413 cm^−1^ found in the Raman spectrum of sample K-4 confirmed the presence of the basic copper carbonate azurite. Raman bands at 620, 650, 1000, and 1040 cm^−1^ in the spectrum of sample F-1 confirmed the presence of the basic lead carbonate hydrocerussite and bands at 1014, 1454, 1580, and 1599 cm^−1^ enabled the confirmation of indigo in sample F-1 [41].

Table 2 summarizes the compounds identified by FTIR spectroscopy, indicating the IR bands of analytical interest and the assignment to functional groups. The remaining IR spectra acquired in the grounds and paint layers in the samples are provided in Appendix A.

Figure 5 shows three IR absorption spectra illustrating the main features of analytical interest found. Figure 5a shows the IR spectrum obtained in the ground of sample J-1. The intense bands at 1406 and 1027 cm^−1^ which are ascribed to lead white, calcite, and clayey minerals dominate the spectrum. Calcite and clayey minerals are soiling materials deposited during the aging of the panels. These minerals, and eventually dolomite, are also present in most of the samples as impurities of the ground gypsum. They have been identified by their characteristic bands: calcite, ν_3_ stretch at 1406 and ν_2_ and ν_4_ deformation at 876 cm^−1^ and 710 cm^−1^ (see Figure 5d, Appendix A); dolomite, ν_3_ stretch 1439, 1399 cm^−1^ and ν_2_ and ν_4_ deformation at 876 and 728 cm^−1^ (see Figure 5d, Appendix A); clayey minerals, Si-O stretch at 1027 cm^−1^ (see Appendix A) [42].

Azurite was identified in sample K-4 by the carbonate ν_3_ stretch at 1461 and 1402 cm^−1^ and the carbonate ν_4_ deformation at 771, 746 cm^−1^ (see Figure 5c, Appendix A) [34].

Identification of calcium sulfate minerals used in the grounds was carried out using FESEM-EDX. The X-ray spectra showed characteristic emission lines of Ca and S. The prevalence of gypsum, of the calcium sulfate dihydrate crystalline variety, was confirmed by the IR absorption bands ascribed to the stretching vibration of OH groups at 3490 and 3395 cm^−1^, bands at 1680 and 1621 cm^−1^ associated with deformation vibrations of OH groups, ν_3_ stretching vibration bands of sulfate groups at 1100 cm^−1^, and ν_4_ deformation vibration bands of sulfate groups at 667 and 596 cm^−1^ (see Figure 5d, Appendix A) [42]. Apart from the gypsum bands that dominate the IR spectra of grounds, some dolomite and calcite are also identified. 

### 2.3. Organic Components

#### 2.3.1. GC-MS

Separated chromatograms were performed in the grounds and paint layers of the samples. Nevertheless, it only obtained conclusive results in C-4 ground. That was associated with restrictions in sample amount and the severe alterations undergone by the organic components of the paintings (*vide infra*). Figure 6 shows the chromatogram obtained in sample C-4. Table 3 summarizes the percentage of amino acids obtained in the samples analyzed and the values obtained in porcine and bovine gelatins used as reference materials. The occurrence of the amino acid hydroxyproline confirms the use of animal glue as the binding medium of the gypsum grounds. Nevertheless, the amino acid composition obtained in the sample does not fit with that found in the reference materials, which suggests that this proteinaceous medium may have undergone some alteration process during aging.

#### 2.3.2. FTIR Spectroscopy

In contrast to GC-MS, IR spectra acquired in the studied samples provided much information relative to the organic materials used in the paintings and their alteration processes. Table 2 includes the main findings obtained.

Egg yolk used as the binding medium of the pigments is identified by the methyl and methylene stretching vibrations at 2922 and 2852 cm^−1^ that occur at wavenumbers slightly lower than those of animal glue (2227, 2862 cm^−1^) and by the amide I band in the 1700–1600 cm^−1^ region (80% C=O stretch). The amide I vibration is hardly affected by the nature of the side chain. It depends, however, on the backbone secondary structure and is, therefore, the amide vibration most commonly used for secondary structure analysis. The SFD-deconvolution process applied to the band has enabled the identification of the overlapped individual bands that can be associated with the main conformations or secondary structures that compose the protein molecules. According to seminal literature [47,48,49,50,51], the wavenumber range of the α-helix conformation is 1648–1657 cm^−1^, β-sheets 1623–1641 and 1670–1695 (weak) cm^−1^, turns 1657–1686 cm^−1^, and random coils 1640–1651 cm^−1^. Figure 5b shows this region of the IR spectrum of sample J-2. Here, the maxima of the α-helix and β-sheets bands at 1651 cm^−1^ and 1623, 1681 cm^−1^, respectively, have been identified. In the 1500–1600 cm^−1^ occurs the amide II band that has a 60% N–H bend contribution and 40% C–N stretch contribution. Amide II shows a low correlation between the secondary structure and frequency and is scarcely used for characterizing the secondary structure of the protein [48,51]. Absorption bands corresponding to amino acid residues and their metal complexes, formed with metal ions released from the pigments, also occur in this region. By applying the SFD process in the IR spectra of the paint layers, a band at 1562 cm^−1^ can be recognized, which is ascribed to the asymmetric stretch of Ca-carboxylate complexes (mainly glutamate and aspartate residues) in the egg proteins. The weak bands at 2550, 1716–1710, and 1595–1510 cm^−1^ are associated with the stretching vibrations of diverse amino acid residues [52,53]. Other interesting features at 1584 and 1538–42 cm^−1^ are observed in Figure 5b. These bands, according to the literature, correspond to Cu, Ca, and Pb-carboxylate complexes formed with free saturated long-chain fatty acids from egg yolk [43,54]. The plateau at 1590 cm^−1^ has been ascribed to the ionomeric species associated with metal soaps formed from drying oils and the lipid matter in the egg yolk [55,56]. Figure 5c shows the IR spectrum acquired in the paint layers of the sample K-4. The IR spectrum is dominated by the bands ascribed to gypsum at 3484 and 3398 (-OH stretch), 1682 and 1619 (-OH deformation), 1095 (sulfate ν_3_ stretch), and 667 and 594 cm^−1^ (sulfate ν_4_ deformation). Calcium oxalate monohydrate (naturally occurring as whewellite) is identified by its characteristic sharp bands at 1619, 1321, and 779 cm^−1^ [57]. Additionally, bands ascribed to egg yolk and azurite are identified. Finally, Figure 5d illustrates the spectral profile of the gypsum grounds of the paintings. Apart from the gypsum bands that dominate the IR spectrum, calcium oxalate monohydrate is identified.

#### 2.3.3. Optical Microscopy

Examination of cross-sections and thin sections of the wood support has enabled the identification of the botanical family. Figure 7 shows the microphotographs of the wood support. Figure 7a shows the cross-section of the support of sample D-1. The boundary between two growth rings is observed in the center of the image. Two tube-shaped intercellular resin canals are placed close to it, characteristic of Pinaceae trees. Figure 7b shows a typical tracheid pitting of Pinaceae families in the radial section [58].

Examination of the surface of the samples has enabled the identification of hyphae (Figure 8a–c) and spores (Figure 8a–d) of fungi that have colonized the paintings.

#### 2.3.4. Surface Analysis

The use of FIB for performing trenches on the surface of the paintings has provided morphological and chemical information in combination with the FESEM-EDX instrument. Figure 9a shows the trench obtained with the FIB in sample F-1. A protective layer of varnish lies on the paint layers. This technique provides a smooth cross-section that characterizes the morphology of the lead white grains used in layer two of sample F-1 (Figure 9b). The chemical composition of the pigment was confirmed using X-ray microanalysis. The image shows aggregates and individual grains of pigment that exhibit a varied particle size distribution that ranges between very coarse (>40 µm) to very fine (<1 µm). The aggregates have rounded shapes, whereas individual grains exhibit the elongated cross-section of their hexagonal tabular habit [15,33].

Figure 10a shows the secondary electron image and the elemental composition obtained in the upper part of the organic protective coating of sample K-4. The surface of paint samples C-2, C-5, and D-1 has shown similar morphology, as shown in Appendix A. The surface is irregular due to the dense network formed by the hyphae of the fungi that colonize the paintings. Atmospheric particles of pollutants and compounds formed due to the alteration processes also increase the roughness of the surface. The chemical composition of the varnish in several points (1–5) in Figure 10a is provided in the homologous spectra shown in Figure 10b. X-ray spectrum 1corresponds to the composition of the point 1 in the bottom of the trench at a depth (at *ca.* 10 μm) is provided in Figure 10b (X-ray spectrum 1). A high emission line of C occurs in the spectrum together with those of Pb and Ca. The amorphous morphology of the varnish is severely altered by the fungi whose hyphae (see (*) mark in Figure 10) infiltrate the varnish and generate pores with a size of a few µm connected across fissures. The latter promote the diffusion of external species (Cl, P) that seem to have precipitated and covered the internal surface of the pores and fissures (spectrum 2) or formed new phases in the varnish (spectra 3 and 4). The varnish composition obtained at the top of the layer (spectrum 5) suggests an enrichment in Ca and Mg.

The trench performed in sample J-4, shown in Figure 11a, illustrates the high degree of alteration of the varnish in the upper 10 μm (see Figure 11b which shows the sample before performing the trench). The surface of the varnish is widely fragmented, and the cross-section shows a dense network of interconnected pores of irregular shape and size. Similar to the other samples, fungi hyphae are present in this alteration layer. The chemical composition of the varnish is illustrated by the X-ray spectrum 1, shown in Figure 11c. A high emission line of C occurs in the spectrum together with those of Pb (38.6%) and Ca (22.98%) and, in a lesser percentage, with Mg (3.03%), Al (3.07%), Si (3.95%), K (2.56%).

#### 2.3.5. Mechanical Properties

Atomic force microscopy-nanoindentation (AFM-NI) has provided helpful information about the mechanical properties of the panel paintings. The advantage of this technique is the minimum amount of sample required in contrast with the conventional techniques for tensile testing in which the specimens of several cm^3^ are destroyed during the trial. Figure 12 shows the AFM topographic maps acquired in peak force error mode in the top varnish layer (Figure 12a), middle cinnabar layer (Figure 12b), and bottom gypsum ground (Figure 12c) of sample C-2. The average elastic modulus (Young’s modulus) values obtained in each layer are 4.4 ± 0.7 GPa, 0.74 ± 0.03 GPa, and 0.86 ± 0.05 GPa, respectively.

## 3. Discussion

The materials and artistic technique as well as the alteration processes undergone by the polychromed panels are discussed in this section in light of the experimental findings.

### 3.1. Materials and Artistic Technique

The analytical results allow the identification of the artist’s palette used in the *alfarje*. The palette comprises six primary pigments: lead white (2PbCO_3_·PB(OH)_2_); cinnabar (HgS); orpiment (As_2_S_3_); azurite (Cu_3_(CO_3_)_2_(OH)_2_*)*; indigo (C_16_H_10_N_2_O_2_), and gypsum (Ca(SO_4_)·2H_2_O) as inert material for the grounds. These pigments are part of broader palettes identified in Hispano-Muslim monuments and architectural elements of a similar chronology [11]. The artistic technique of the Palma de Mallorca panel paintings is characterized by the use of pure pigments applied in successive paint layers, starting with a pigment used as a base. In this case, this seems to be the red cinnabar, which appears in most samples as the bottom layer.

It has been reported that using pigments in the Islamic world or geographic areas with Islamic influence (*Mudéjar* art) was quite common. The taste for bright and contrasting tones was popular, so high-quality pigments were often used. Some significant examples can be cited that confirm this: Qusayr ‘Amra in Jordan (Umayyad wall paintings, 8th century) [59]; in the Nasrid period (14th century), the Alhambra [20,22] or the Dar al-Manjarra al Qubra [19]; and in *Mudéjar* architecture the facade of the palace of King Pedro I [27,28] or the plasterworks and the *alfarje* of Courtyard of the Maidens, both in the Royal Alcázar of Seville, built between 1356–1366 [29]. The color palette used by these artisans consisted of red, green, yellow, blue, black, and white hues. They used pigments such as cinnabar, vermilion, iron oxides, or minium to make the reds. The green hues were made with malachite and verdigris, which were sometimes mixed with yellow pigments such as yellow ochre, orpiment, or even the lead and antimony yellow, which Muslim artists used from very early stages for their use in ceramic decoration [59,60]. The color blue was usually made with azurite or natural ultramarine blue [11]. Although it has been reported that most of the azurite pigment used by Spanish painters in the Middle Ages came from Germany, the local needs were probably covered with minerals obtained from the Iberian Peninsula, where copper sources can be found in almost all regions. Thus, the azurite identified in the *alfarje* was probably the Spanish *azul fino* (fine blue) obtained from local sources in the Iberian Peninsula rather than the most expensive *azul de Alemania* (German blue) [61]. The orpiment was used for the yellows, common in Hispano-Muslim and *Mudéjar* art [11,26]. In the medieval Islamic world, orpiment was commonly applied for making manuscript inks and coloring architectural elements [62,63]. Cinnabar was the pigment for red colors. This pigment can be artificially prepared or by simply grinding the mineral cinnabar. There are three local sources of cinnabar mineral in Ibiza, one of the Balearic Islands. In particular, in S’Argentera (Puig De S’Argentera), Sant Carles De Peralta, and Santa Eulària Des Riu (mindat Locality ID: 290899). In the Iberian Peninsula, the largest source of cinnabar in the world were the mines of Almadén (Locality ID: 3115) [40,64]; thus, Hispano-Muslim artists had easy access to this raw material [27]. Charcoal black and bone black were employed for decorating in black [11]. White lead was used in panel painting, whereas gypsum plaster mixed with paste lime was employed in plasterworks [11]. This mixture would retard the setting of the plaster, provide whiteness, and improve its resistance to humidity.

Indigo was widely used in the Islamic world in manuscripts due to its high coloring power and low economic cost [62,65,66,67]. The use of this pigment in the wooden ceiling that decorates the dome of the Al-Aqsa Mosque in Jerusalem, dated back to 1345–1350, is one of the scarce applications reported in the literature [68]. It has also been identified in Casa Pilatos (Seville, *Mudéjar*), where it was employed together with a yellow resin to obtain a green color [26]. The use of indigo by medieval European painters for decorating panel paintings was also rare. The only report is a Norwegian church’s painted wooden altar frontal [36]. Therefore, using indigo as a blue pigment in the Palma de Mallorca panel paintings seems to be an unusual procedure.

Lead white was the most common white and has been used since antiquity [29]. Hispano-Muslim artists used lead white alone or mixed in some pictorial layers, such as the one where indigo is identified [68,69]. Interestingly, the Italian artist Cennini describes the use of combinations of white pigments with indigo in his art treatise [70]. 

Regarding the preparations that serve as the basis for polychromies on wood, different materials are identified that provide color and consistency to this layer in Hispano-Muslim and *Mudéjar* art, the most frequent being white (gypsum or with a mixture of plaster and calcium carbonate) but also orange or red, made with red lead, or even yellow, made with orpiment. The choice of one hue or another was related to the predominant tones in the paintings [11].

The ground has been prepared according to the medieval common practice of *gesso* in which the gypsum is applied in one or several successive layers and the ground is finished off with a final layer of animal glue [70]. This binding medium reduces the absorption of the *gesso* porous surface. An innermost layer of an oil sizing was identified by means of microchemical tests (not presented analyses). This material is used as a sealant coating for better applying upper layers of ground with inert materials such as gypsum [71]. Nevertheless, this ground exhibits a notable variation in thickness between the samples analyzed. Thus, in C-5 and D-1 samples, gypsum is only found infiltrated in the internal channels of the wood cells of the support so that paint layers seem to be directly applied on the wood support. 

The use of egg yolk in the paint layers of the *alfarje* panels, as suggested by the identification of proteins in these layers by FTIR spectroscopy, confirms the technique of tempera *grassa*. Using animal glue as a binding medium in the ground is unambiguously confirmed by identifying proteins containing hydroxyproline. 

Natural resins have been used mainly as varnishes since antiquity. They were often mixed with oils or waxes to improve their toughness or adhesivity and increase their melting point [63,72,73]. In the eleventh century, natural resins began to be melted in oils and applied hot, which provided rough surfaces [63]. Linseed oil and diterpenic resins have been identified in the Alhambra complex [20,21], in the oratory of the Yusuf I Madrasa, Granada [74], or the *Mudéjar* façade of King Pedro I, Seville [27,28]. 

The analyses carried out in the Palma de Mallorca panel paintings were unsuccessful in the identification of the varnish used as protective coating, due to the limited disposal of samples, the elusive character of these materials, and the occurrence of severe alteration processes of chemical and microbiological origin (vide infra). Nevertheless, some clues are provided by the IR spectrum of sample C-4 which shows methyl and methylene IR bands characteristic of terpenoid resins, together with those ascribed to a drying oil. It suggests that the varnish could be prepared with this natural resin dissolved in a drying oil.

Remarkably, the use of thin foils of silver covered with a varnish layer changes the color of the metal, simulating gold in sample J-4, or the identification of a thick layer of tin that underlies a white and blue paint layer. This artistic technique has also been identified in the Alhambra, where a tin leaf is covered by an organic layer of chestnut tone [20]. The most common technique for making the metal foil adhere to the underlying surface is the application of a thin layer of finely powdered earth pigment (bole) beneath. The pigment is bound with a proteinaceous glue, which acts as an adhesive for the foil. In the Nasrid dynasty of the Hispano-Muslim period, the use of gold sheets on thin layers made with bole (earth pigment) was common [11,20,25,75], while, in *Mudéjar* art, for example, in the Alcázar of Seville, gold is applied with *mixtion* (drying oil with siccative pigment) [27,28]. Other materials, besides gypsum, were found just underneath the metal foils. On the other hand, weak emission lines of Pb are present in the X-ray spectrum acquired in this area. Lead pigments, such as lead white or litharge, were added to drying oils in small amounts as siccative of drying oils [11,26]. This finding suggests that a *mixtion* could be used as an adhesive in the Palma de Mallorca panel paintings for fixing the metal foils over the gypsum ground [76,77].

### 3.2. Alteration Processes

A variety of alteration processes, some of them acting simultaneously, have been identified.

#### 3.2.1. Pigments and Metal Foils

The X-ray spectra of the thin silver leaf in sample J-4 (see Appendix A) show an intense sulfur emission line that accompanies silver lines. The Ag/S molar ratio value of 1.8 is near the ideal stoichiometric proportion of 2.0, which corresponds to silver sulfide (Ag_2_S, mineral acanthite) [40,78]. The formation of silver sulfide under atmospheric conditions, often called tarnishing, is a frequent corrosion process exhibited by silver alloys and silver objects in air environments with high levels of sulfur pollutants, such as SO_2_. pH, humidity, temperature, pollutants, and exposure time influence that process [78]. The identification of gypsum formed on the surface of the paintings (i.e., see Figure 5c) confirms the alteration activity of the atmospheric S-containing pollutants.

The thick tin leaf found in sample J-2 also showed signs of deterioration, as seen in Figure 13. The metal in contact with the gypsum ground is entirely altered, forming a mass of tabular crystals. The central part of the lead has also transformed its compact metallic structure into a cross-linked network of needle and tabular crystals. These morphologies resemble those of Sn(IV) oxide (SnO_2_, cassiterite) and Sn(II) oxide (SnO, romarchite). Formation of these tin oxides as corrosion products of tin leaves applied in French medieval stone sculptures decorated with the technique of tin-relief brocade has been recently reported [79].

Another interesting finding concerns the visual appearance of the indigo pigment used in the paintings. The greenish hue acquired by indigo aggregates indicates some oxidation process acting on this pigment. Similar transformations when indigo is used in oil paintings have been described. Free radicals formed in the drying oil used as a binding medium due to aging can act as catalysts for the oxidation of pigments such as indigo or Prussian blue. In particular, the indigo molecules are oxidized to the orange isatin, as illustrated in Figure 1 [80]. Another oxidation process of indigo involves the Maya blue preparade by the ancient Maya artists. This pigment has become famous due to its particular nanostructured hybrid organic–inorganic composition. The greenish hue achieved in some varieties of Maya blue is due to redox-tuning processes undergone by indigo. The blue indigo attached to the internal channels of the palygorskite clay oxidizes to dehydroindigo or isatin during the moderate heating process to which the organic pigment was subjected as part of its preparation procedure. The higher temperature, the more significant amount of indigo reacted. Thus, a variable amount of indigo molecules could oxidize to the yellow dehydroindigo or orange isatin molecules in a solid-phase catalyzed reaction so that the resulting mixture of indigo, dehydroindigo, and isatin exhibits a greenish color (Figure 1) [81]. The metabolic activity of the fungi occurring mainly in the Palma de Mallorca paintings is a plausible cause of the color change observed in the indigo pigment. Another probable cause of the alteration of indigo, in combination with the oxidative ability of fungi, is the photochemical formation of free radicals from the altered lipid molecules of the binding media that also should act as catalysts of the oxidation process.

#### 3.2.2. Binding Media and Varnishes

As described in the prior section, egg yolk alone or emulsified with a drying oil could be used as the binding medium in the paintings. In addition, a drying oil could also be used for preparing the protective varnish and adhesive for attaching the metal sheets to the ground. Some alteration processes affecting these materials have been recognized using the FTIR technique. Partial hydrolysis of the triacylglycerols (TAGs) of the lipid binders is identified by the absence of the characteristic carbonyl stretch of ester groups (1738 cm^−1^) and the occurrence of carbonyl group stretch in free fatty acids (1703-08 cm^−1^) (see Appendix A) [82,83]. Hydrolysis takes place after the TAGs form the cross-linked molecular network during the autoxidation and polymerization stages of the drying. Further natural aging involves spontaneous hydrolysis of TAGs, preferably close to the surface of the pigment grains. The latter should act as Lewis acids that disturb the CO links in the TAGs [84]. Macroscopically, the hydrolysis reaction causes a loss of the mechanical properties in the paintings [83]. The exposition to light, humid environments, or high-temperature processes are adverse conservation conditions that promote the formation of metal soaps, carboxylic metal complexes formed from long-chain fatty acids, and some pigments prone to form metal soaps. These compounds occur mainly in oil paintings and are preferably formed from pigments used as siccatives [85]. Less frequently, they are found when egg yolk is used as the binding medium [44]. The formation of metal soaps has been pointed out as responsible for several painting damages, crusts, efflorescences, hazes, cracks, detachments, and loss of mechanical resistance [86,87,88]. Three types of carboxylic metal complexes have been identified [89]. During the autoxidative drying of the binder, some TAG molecules near the pigment grains can acquire some carboxylic groups and coordinate with the metal ions on the surface of the pigment grain. This first type remained attached to the polymeric network of binder and were adsorbed into the pigment surface. The metal ions released from the pigment migrate into the polymerized network linking to carboxylic centers and forming an ionomeric phase. A polymeric cross-linked structure is now formed by links between carboxylic centers in which the metal ions act as bridges [56]. Due to the aging progress, the hydrolysis process is ongoing, and the divalent metal ions, which still have the disposal of a second bonding center, can react with the new free fatty acids. Then, the network achieves a metastable state due to the supersaturation of metal soaps that spontaneously crystallize [89]. Three metal soaps have been unambiguously identified in the Palma de Mallorca paintings: Ca-carboxylic complexes (1576, 1539 cm^−1^), Pb-carboxylic complexes (1541, 1517 cm^−1^), and Cu-carboxylic complexes (1584 cm^−1^). Three aspects that confer singularity to this finding are remarkable. Ca-carboxylic complexes and, to a lesser extent, Pb- carboxylic complexes have been identified in the ground, suggesting that migration of free fatty acids from upper layers occurred. The broad band at 1500–1590 cm^−1^ suggests that the amorphous ionomeric phase is prevalent over the crystalline forms of metal soaps. This is confirmed by the absence of nodules of metal soaps supported by the FESEM-EDX examination. The identification of bands ascribed to Ca-glutamate and Ca-aspartate complexes is a piece of evidence that the proteins of the egg are contributing to the ionomeric phase.

Ca-oxalate monohydrate has been unambiguously identified in the paintings by its characteristic features in the IR spectra of grounds and paint layers. The exact origin and mechanism of the formation of oxalate salts in paintings, monuments, and archaeological sites still need to be completely elucidated. Nevertheless, many researchers propose two main causes: the metabolic activity of microorganisms and the presence of organic materials in aggressive environments [90,91,92,93,94]. In the Palma de Mallorca paintings, the confluence of the two factors is achieved. Therefore, a complex mechanism involving physicochemical factors and the metabolic activity of fungi has caused the formation of large amounts of this compound. Interestingly, the Ca-oxalate is sometimes accompanied by Pb-oxalate and, eventually, by Cu-oxalate. The latter are discerned from Ca-oxalate by the stretching vibrations of the carboxylate group at 1660, 1365 cm^−1^ (Pb), and 1643, 1360 cm^−1^ (Cu) [45,46]. The occurrence of these compounds demonstrates the high degree of alteration of the organic matter that has significantly hindered the identification of the raw materials used by the artist as binding media and a protective coating.

Similar to the binders, the raw materials used for preparing the varnish could not be accurately identified by chemical analysis due to their high degree of alteration. Figure 14 shows the backscattered electron image of the upper layers of sample C-4. It shows abundant grains of varied shapes, sizes, and compositions scattered in the matrix of the varnish illustrating the diversity of compounds in this outer layer. Pb and Ca are spread almost everywhere in the matrix layer. That wide distribution is associated with the ionomeric phase of carboxylic complexes and crystalline oxalates. Clusters of columnar microcrystals are dispersed and formed near the fissures that penetrate the varnish. The K/S molar ratio value of 2.08 found for these compounds suggests adscription to potassium sulfate monohydrate. This salt, known as the mineral syngenite, is commonly found in altered monuments. Chloride salts cover the internal surface of fissures. The varnish accumulates calcite and clayey minerals on the surface. These minerals sometimes also infiltrate the varnish, and eventually, apatite minerals are also found.

This heterogeneous composition resembles that of a paint layer. That, together with the high degree of polymerization that terpenoid varnishes achieve on aging, is the cause of the varnish’s higher stiffness compared to those obtained in the layers underneath. Figure 15 shows, as an example, an EM linescan graph obtained in the varnish layer of sample C-2. This graph illustrates the heterogeneity of the varnish layer. Null values of EM are found in fissures and pores, while a variety of EMs that achieve values of 20 GPa are measured in the mineral and organic phases placed along the line. These values agree with those reported in the literature for 19th century oil paintings, with average EM in the range of 4–13.6 GPa [95]. On the other hand, the lower EM values found in the paint layer and ground are associated with the use of proteinaceous binding media.

#### 3.2.3. Biodeterioration

The presence of fungi has significantly damaged the Palma de Mallorca paintings by triggering a dense network of hyphae and spores that cover and interpenetrate the upper layer of the painting. The more drastic alteration in brightness and roughness of the surface is observed in the upper 8–10 µm outer layer, as was shown in the secondary electron images and X-ray spectra acquired in the trenches (see Figure 7, Figure 8, Figure 9 and Figure 10 and Appendix A). Nevertheless, the colony’s growth has also resulted in more severe alterations. One of the more adverse effects induced is the loss of cohesion due to forming of fissures and pores. As mentioned previously, fungi are probably the cause of the formation of oxalate salts. Although no evidence of the direct participation of fungi in the oxidation of indigo has been found, abundant references to the ability of fungi to decolorize dyes can be found in the literature [96]. This capability has been exploited for removing industrial dye residues in wastewater species. One of the fungi most frequently used is *Aspergillus niger* [97], which is likely one type of fungi found in the Palma de Mallorca panel paintings. Recently, a pathway has been proposed for the degradation of the synthetic indigo carmine, a sulfonated derivative of indigo, by the metabolic activity of a fungus species [98]. The sulfonated derivative of isatin and several related compounds were found among the metabolites yielded by the fungus after incubation of the pigment with culture filtration of the microorganism. The proposed mechanism goes by two different pathways that decompose the original indigo carmine molecule in new molecules characterized by (see Figure 2): (1) loss of one of the sulfonate groups and (2) cleavage of the central double bond in the indigo molecule to yield isatin sulfonate derivatives. Thus, it is plausible that fungi that have colonized the Palma de Mallorca paintings have developed a similar metabolic procedure for oxidizing indigo molecules and changing the pigment’s color to a blue-greenish due to the coexistence of blue indigo and orange-yellow isatin molecules.

#### 3.2.4. Implications of the Alteration Processes in the Conservation of the Paintings

The performed study has relevance not only for the archaeometrical information provided but also to help stakeholders and conservators make proper decisions on the best method for preserving the paintings. As one of the few remaining examples of wooden coffered ceilings decorated with paintings of the medieval Hispano-Muslim period, the *alfarje* is a valuable part of pictorial heritage. The identification of altered indigo and metal foils, the loss of mechanical resistance of the paintings, the intense degradation observed in the surface due to the latter, the infiltration of soluble salts, and the mineral deposits are alteration processes that require specific conservation treatments in the short term. 

All conservative and restorative treatments that will be carried out on the panels must follow intervention criteria established by professionals and the scientific community, taking into account the nature and materials that make up the *alfarje* [99]. The interventions must be respectful, and it is necessary to carry out graphic and photographic documentation beforehand. A complete characterization of the state of conservation of the pieces must be based on preliminary scientific analysis and examination of the *alfarje*. Treatments carried out in artworks similar to the *alfarje* must be used as starting point for planning the intervention and selecting methods [100,101,102,103]. Later, tests and trials must be carried out for all stages of the intervention, from the elimination of superficial deposits to the evaluation of the materials used for consolidation.

In addition to immediate intervention, a long-term program for preventive conservation is needed. The latter’s planning requires considering the probable effects that climate change can induce in the *alfarje*. It is well known that changes in humidity or the environmental parameters that control thermal shock or freezing and thawing are critical for conserving paintings. In addition, studies on the effects of climate change on cultural heritage have revealed variations in temperature, rainfall, and frequency of flooding with heavy rain and droughts [104] For this reason, in the last several decades, attention has been paid to developing methods for assessing the vulnerability of cultural goods to the impact of climate change [105]. In particular, paintings and wooden objects are susceptible to loss of mechanical resistance and increased biodeterioration if the historical climate changes. Recently, several studies have been aimed at developing methods for reconstructing historical data and methods for the probabilistic forecast of temperature and humidity over the following centuries that can assist the conservators in the task of preventing new alterations or stopping the aggravation of those already observed [104,106]. Therefore, a specific method for monitoring the environmental conservation conditions of the *alfarje* should be carried out based on the literature background. 

## 4. Materials and Methods

### 4.1. Description of the Polychromies and Sampling Strategy

Figure 16 shows some wooden panel paintings in the Palma de Mallorca *casal*. In some of the panels, decoration motifs are recognized, such as panel D, which shows calligraphic signs, or panel F, which presents floral ornaments. Sampling was carried out at different points of those panels following 2 main criteria: (i) identification of the artist’s palette, and (ii) characterization of alterations. Samples were carefully taken with a scalpel due to the poor state of conservation observed in the panels. In panel C, 5 samples were taken; in panel D, 2 samples were taken; in panel F, 4 samples were taken; in panel J, 4 samples were taken; and in panel K, 4 samples were taken. 

### 4.2. Samples Analyzed

Ten samples, which consisted of microfragments of the polychromies that include all the layers, have been analyzed. Three of them, labeled C2, C4, and C5, were excised from panel C, one sample labeled D1 from panel D, two samples labeled F1 and F4 from panel F, three samples labeled J2, J3 and J4 from panel J, and one sample labeled K4 from panel K.

### 4.3. Instrumentation

The multi-technique strategy was carried out using the following instrumental techniques:Optical microscopy: Stereoscopic microscope Leica GZ6 (X10-X50) and Leica DM2500 P (Leica Microsystems, Heidelberg, Germany) were used for preparing and examining cross-section samples, respectively. IC80HD Leica Digital FireWire Camera (DFC) (Leica Microsystems, Heidelberg, Germany) with Leica Application Suite 4 (LAS) software was used for acquiring and processing the digital images.FESEM-EDX: Zeiss model ULTRA 55 (Carl Zeiss AG, Oberkochen, Germany) that operates with an accelerating voltage of 0.8 kV in the electron source and working distance of the detector 2.9–4.0 mm.FIB-FESEM-EDX: Zeiss (Carl Zeiss AG, Oberkochen, Germany) model Auriga compact instrument operating at 30 kV and 500 µA. For sectioning samples and forming the trenches the FIB was operated at 20 nA for generating the focused beam of Ga ions. By tilting the stage 54° with the microsample, the Ga beam could be oriented perpendicularly to the plane of the vertical wall of the trench. The secondary and backscattered electron images were acquired at 2 kV in the FESEM. X-ray spot analyses were performed in trenches with an Oxford-X Max X-ray microanalysis system (Oxford Instruments, Abindong, UK) controlled by Aztec software (Oxford Instruments, Abindong, UK). A voltage of 20 kV and a working distance of 6–7 mm was used.GC-MS: Agilent 5973N mass spectrometer with an Agilent 6890N gas chromatograph (Agilent Instruments, CA, USA) operated with Agilent Chemstation software (MSD) for the integration of peaks and the evaluation of mass spectra. The chromatographic separation was achieved with a chemically bonded fused-silica capillary column HP-5-MS (Agilent Technologies, CA, USA) (stationary phase 5% phenyle 95% methylpolysiloxane, 30 m, 0.25 mm i.d., 0.25 µm film thickness). Chromatographic conditions: initial temperature 100 °C followed by a gradient of 5 °C min^−1^ up to 155 °C, ramping up at 15 °C min^−1^ up to 295 °C, which were held for 5 min. He carrier gas was used with inlet pressure of 99.89 kPa and 1:20 split ratio. Electronic pressure control was set to constant flow mode (1.3 mL min^−1^) with vacuum compensation. Ions were generated by electron ionization (70 eV) in the ionization chamber of the mass spectrometer, which is scanned in the range m/z 20 to 800, with a cycle time of 1 s.FTIR spectroscopy: Vertex 70 Fourier-transform infrared spectrometer (Bruker Optik GmbH 2012, Ettlingen, Germany), equipped with a FR-DTGS (fast recovery deuterated triglycine sulfate) temperature-stabilized coated detector and a MKII Golden Gate Attenuated Total Reflectance (ATR) accessory (Specac Ltd, Orpington, UK). The IR spectra were acquired by collecting 32 scans at a resolution of 4 cm^−1^. Spectral range scanned was 500–4000 cm^−1^. The spectra were processed using the OPUS 7.2/IR software (Bruker Optik GmbH 2012, Ettlingen, Germany).μRaman spectroscopy: Confocal Raman Microscope Xplora MTB model supplied by Horiba Scientific, Kioto, Japan, that operates with laser beams of 532 nm or 738 nm as excitation source with a maximum power of 90 mW. Other operating conditions were: samples acquired with backscattering geometry at room temperature; microscope objectives x100 and x10 confocal for focusing the excitation laser on the sample and collecting the scattered light to the spectrometer; measurements were performed in 6 different areas per sample to obtain representative results. Exposure time, number of acquisitions, and laser power varied between 10^−1^ s, 10^−5^, and 30–80 mW, respectively, and a grating of 1800 groves mm^−1^ was used in all the measurements. The instrument was controlled with a LabSpect 6 Spectroscopy Suite from Horiba MTB software (Horiba Scientific, Kioto, Japan).AFM-NI: The determination of the elastic modulus (EM) was carried out in the microsamples of the original polychromies prepared as cross-sections. A multimode AFM (Digital Instruments VEECO Methodology Group, New York, NY, USA) with a NanoScope IIa controller was used. This instrument is equipped with a J-type scanner with a maximum scan size of (150 × 150 × 6) mm. EM in strata of interest in selected cross-sections was obtained in the ScanAsyst peak-force quantitative nanomechanical mode (QNM) with a tip RTESPA-300 (Bruker Optik GmbH 2012, Ettlingen, Germany) (k = 40 N m^−1^). Calculation of EM values was carried out with Bruker Nanoscope 1.40 Analysis software. Images created for each paint layer during one scan with scan sizes of (20 × 20) and (40 × 40) µm consisted of 256 lines by pixels and were taken at a scan rate of 0.4–0.5 Hz.

### 4.4. Reagents

The following reagents were used to treat the samples prior to the GC-MS analysis: methyl chloroformate (ECF) (purity 99%) (Sigma-Aldrich, Steinheim, Germany); absolute pyridine and chloroform at 98% for GC (Acros, Cambridge, MA, USA); absolute ethanol (Carlo Erba, Rodano, Italy); sodium hydrogen carbonate (Panreac, Barcelona, Spain).

### 4.5. Preparation of Samples for Analysis

Samples that consisted of microfragments of the paintings were firstly examined with optical microscope. After this, selected microfragments were prepared following 2 different procedures: (a) embedding the microsample of polychromy in polyester resin and polishing with abrasive dishes of SiC until a uniform cross-section was obtained. Polyester resin (Glasspol 328, Glasspol Composites SL, Valencia, Spain), which does not require heating for curing, was used and for excising pigments from the different layers with a thin needle. These were then dispersed on a glass slide for examining in the petrographic microscope.

Microfragments and cross-sections previously examined by optical microscopy were mounted on aluminum disks adapted to be placed in the vacuum chamber of the FESEM-EDX. All the analyzed samples were carbon-coated to avoid localized charging and distortion or reflection of the electron beam. 

Trenches were obtained with the FIB on the microfragments previously examined and analyzed with the optical microscope and FESEM-EDX. The microsamples were fixed with carbon adhesive to the aluminum stage of the FESEM. The size of the trenches was (10 × 8 µm). Trenches were performed at points on the surface of the paint microsamples where cracks, pits, fissures, or mineral microdeposits were not present. 

For analyzing the samples in the GC-MS, a small quantity (0.5 mg) of sample was ground and placed in a 0.3 mL minivial (Supelco Bellefonte, PA, USA). Hydrolysis was carried out with 100 μL of 6 M HCl in an N_2_ atmosphere for 24 h at 110 °C. The hydrolysate solution was gently evaporated to dryness under an N_2_ flow. Following this, 100 μL of water and 100 μL of CHCl_3_ were added and shaken vigorously to facilitate the extraction of fatty acids in the chloroformic phase so that the aqueous and chloroformic phases were separated and treated independently. A total of 50 μL of the aqueous phase containing the proteinaceous components was dissolved in 50 μL of ethanol/pyridine 4:1. The resulting solution was treated with 8 μL of ECF for 10 min and extracted with 50 μL of 1% CHCl_3_ in ECF. Afterwards, 50 μL of a saturated solution of NaHCO_3_ was added. A total of 1 μL of the chloroformic phase was injected into the chromatograph.

Samples of paint and ground layers were excised with the help of a scalpel and directly exposed to the ATR window of the FTIR spectroscope.

Cross-sections of polychromies were directly exposed to the laser beam in the μRaman spectroscope and AFM probe.

### 4.6. Processing of Spectra and Topographic Maps


X-ray spectra acquired with the FESEM-EDX and FIB-FESEM-EDX instruments were processed using the ZAF method. The counting time for acquiring X-ray spectra in quantitative analysis was 100 s. More details relative to the accuracy and repeatability of quantitative measurements in the working conditions can be found in Appendix A.FTIR spectroscopy: information about the secondary structure of proteinaceous binding media has been obtained in the region of amide I using the Fourier self-deconvolution (FSD) FSD-deconvolution resolution-enhancement technique [47]. Band width and resolution enhancement factor parameters applied in the mathematical process were in the range of 18–19 and 0.65–0.70, respectively. The spectra were processed using the OPUS 7.2/IR software.The values of the reduced elastic modulus (Er) were obtained for each pixel by mathematically fitting the retract curve region [107] using the Derjaguin, Muller, Toropov (DMT) model [108], which established the equation:(1)$$F_{\mathrm{tip}}=\frac{4}{3}E_{r}\sqrt{Ra^{3}}+F_{\mathrm{adh}}$$
where *F*_tip_ and *F*_adh_ are the force on the tip and the adhesion force *d* is the tip-sample separation and *R* is the radius of the tip end.


On the other hand, the geometry of the indenter must be characterized to obtain accurate results. The tip radius R is experimentally determined by an indirect method that determines the cantilever’s spring constant by performing a thermal tune and deflection by pressing the tip onto a sapphire disc. A polystyrene sample with a known EM of 2.7 GPa was used.

Following this, the EM of each layer is calculated from reduced modulus *E*_r_, which is given by the expression:(2)$$E_{r}={\left[\frac{1-\nu_{t}{}^{2}}{E_{t}}+\frac{1-\nu_{s}{}^{2}}{E_{s}}\right]}^{-1}$$
where *ν*_t_ and *ν*_s_ are the Poisson’s ratio of the tip and sample layer, respectively, and *E*_t_ and *E*_s_ are the EM of the tip and sample layer, respectively. In this mathematical procedure, the tip modulus *E*_t_ is assumed to be much larger than the sample modulus *E*_s_ and, therefore, the former came close to infinite. Then, Equation (2) can be reduced to: (3)$$E_{r}={\left[\frac{1-\nu_{s}{}^{2}}{E_{s}}\right]}^{-1}$$

It has been found that the elastic modulus *E*_s_ generally ranges between about 0.2 and 0.5 (the latter value for perfectly incompressible materials). A value of 0.3, which is recommended for materials with an EM that falls within the 1 GPa < *E*_s_ < 10 GPa range, was used. Each layer’s elastic modulus EM (*E*_s_) can be calculated using the sample Poisson’s ratio. Thus, EM is provided automatically by the instrument for each pixel of the image. EM is calculated as the average value of the individual EM in each pixel of an image. A minimum of 3 images were acquired in each sample layer, which provided a repeatability of at *ca.* 5%. 

## 5. Conclusions

The study carried out has enabled us to conclude:The polychromed wooden panels found in the Palma de Mallorca *casal* are an excellent example of the fusion of cultural influences that took place in the Iberian Peninsula and Balearic Islands. These Hispano-Muslim paintings are made with a ground of gypsum (Ca(SO_4_)·2H_2_O) and animal glue and using egg yolk to bind the pigments in the paint layers. The color palette unambiguously identified is composed of white lead (2PbCO_3_·PB(OH)_2_), cinnabar (HgS), azurite (Cu_3_(CO_3_)_2_(OH)_2_), orpiment (As_2_S_3_), and indigo (C_16_H_10_N_2_O_2_).Although direct analytical evidence of the use of terpenoid resins, egg yolk, and a drying oil by GC-MS could not be obtained due to the significant transformation of these compounds to oxalate salts, a combination of optical microscopy, FIB-FESEM-EDX and FTIR has provided abundant experimental evidence for sustaining those hypotheses.Several alterations have affected and seriously damaged the paintings: fungi growing on the surface in combination with a high humidity level have resulted in the oxidation of metal foils and indigo pigment and the transformation of the organic materials in oxalate salts. This last process affects all the layers of the paintings. Hydrolysis of TAGs and triggering of metal soaps have also been recognized in the paintings. These processes combined have resulted in a loss of the mechanical strength of the paintings.The proposed multi-technique approach has proven to be useful in the characterization of the materials, technique, and alterations of the paintings. In particular, a combination of the advanced FIB-FESEM-EDX and AFM-NI, which are scarcely used in the study of cultural heritage, has been successful for performing surface analysis and for the chemical, morphological, and mechanical characterization of the outer part of the paintings.

## Data Availability

Not applicable.

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
