# Peer review of "New Insights into the Medieval Hispano-Muslim Panel Painting: The Alfarje Found in a Balearic Casal (Spain)"

_molecules, 2023, doi:10.3390/molecules28031235_

Round 1

Reviewer 1 Report

The manuscript of Carla Álvarez-Romero et al. presents results of a comprehensive diagnostic investigation carried out on a medieval polychromed alfarje found in a Hispano-Muslim casal from Palma de Mallorca, Spain. Several multi-layer paint samples were analyzed via a complex analytical methodology that included easy-accessible spectroscopic techniques (FTIR, Raman) as well as more advanced methods – GC-MS, (FIB)FESEM-EDX and AFM-NI. The article is well structured and scientifically sound. The introduction provides a comprehensive background, the results are clearly presented, and the discussion puts into context the main findings. The study provides new insights on the materials and painting techniques in Hispano-Muslim panel painting, indigo being identified among the employed blue pigments.

I have only some minor comments for the authors, as indicated below:

Table 1. I suggest to include another column where to specify for each sample what techniques were used.  Reading the paper, it is not clear if all samples were analyzed with the full methodology.

Figure 1-e. According to Table 1, this sample has 5 layers. However, the 5th layer (the white ground) is not visible in the image provided.

Lines 227-229. Has the presence of orpiment and cinnabar been confirmed via Raman analysis as well?

Lines 232-234. These statements need a reference.

Lines 248-249. Please add a reference.

Section 2.3.2. The authors should move all FTIR data and data discussion related to pigments in section 2.2.2., and leave in this section (2.3.2) only the data related to the organic compounds.

Lines 302-304. Please include in the text the Figure/spectra you are referring too.

Lines 354-355. Please add a reference.

Figure 5-a. I believe that the main peak at approx. 1400 cm-1 can be partially ascribed to the presence of lead white as well.

Lines 507-511. I suggest to remove these lines, the information is too generic.

Lines 565-567. Please add a reference.

Figure 15: Images provided are blurred. Please replace with higher quality images. Also, the sampling locations should be highlighted on the wooden panels.

Section 4.3. For the FTIR analysis, please specify the spectral range as well.

Lines 884-886. Please specify in what software was the FSD-deconvolution carried out.

Author Response

Valencia, 19th January September 2023

Manuscript Number: 2160582

Title: Analytical study of a medieval polychromed alfarje found in a Hispano-Muslim casal (Palma de Mallorca, Spain)

Molecules

Corresponding author: Dr Carla Álvarez-Romero and Prof María-Teresa Doménech-Carbó

Dear Editor

            Thanks for your interesting comments and suggestions. According to them, we have modified the manuscript as follows:

Comments in the review letter:

Reviewer 1

Thanks for your valuable comments and suggestions that undoubtedly lead to improving the manuscript.

Reviewer 1’s comment 1: Table 1. I suggest to include another column where to specify for each sample what techniques were used.  Reading the paper, it is not clear if all samples were analyzed with the full methodology.

According to the referee’s comment, a column has been included in the Table 1 to specify the techniques employed for the characterization for each sample

Reviewer 1’s comment 2: Figure 1-e. According to Table 1, this sample has 5 layers. However, the 5th layer (the white ground) is not visible in the image provided.

Layer 5 corresponding to the ground that can be recognized in the microphotograph in oblique illumination of the sample F1 shown in table S1.

According to the referee’s comment, a footnote has been included in Table 1 that remits the reader to Table S1.

Reviewer 1’s comment 3: Lines 227-229. Has the presence of orpiment and cinnabar been confirmed via Raman analysis as well?

The difficulty for accessing to the MicroRaman instrumentation led the authors to limit the use of this instrument for elucidating the pigments where not was obtained an unambiguously identification with the rest of techniques applied in this study. The MicroRaman analysis of cinnabar and orpiment was considered unnecessary due to the unambiguous identification of the elements As, S and Hg obtained by FESEM-EDX, together with the excellent match of the experimental stoichiometry and that ideal of these minerals, reported in literature, also provided by the quantitative analysis with this technique using the ZAF method for correction of interelemental effects.

According to the referee´s comment, a paragraph has been included in the revised version of the manuscript for clarifying this issue.

Reviewer 1’s comment 4: Lines 232-234. These statements need a reference.

According to the referee’s comment, a quotation has been included in the revised version of the manuscript in this paragraph.

Reviewer 1’s comment 5: Lines 248-249. Please add a reference.

According to the referee’s comment, a quotation has been included in the revised version of the manuscript in this paragraph.

Reviewer 1’s comment 6: Section 2.3.2. The authors should move all FTIR data and data discussion related to pigments in section 2.2.2., and leave in this section (2.3.2) only the data related to the organic compounds.

According to the referee’s comment, data concerning FTIR study of pigments have been moved to section 2.2.2.

Reviewer 1’s comment 7: Lines 302-304. Please include in the text the Figure/spectra you are referring too.

According to the referee’s comment, the text has been rewritten including figure/spectra citations.

Reviewer 1’s comment 8: Lines 354-355. Please add a reference.

According to the referee’s comment, a quotation has been included in the revised version of the manuscript.

Reviewer 1’s comment 9: Figure 5-a. I believe that the main peak at approx. 1400 cm-1 can be partially ascribed to the presence of lead white as well.

According to the referee’s comment, the paragraph has been re-writing for including adscription of 1400 cm-1 band to lead white.

Reviewer 1’s comment 10: Lines 507-511. I suggest to remove these lines, the information is too generic.

According to the referee’s comment the paragraph has been suppressed in the revised version of the manuscript.

Reviewer 1’s comment 11: Lines 565-567. Please add a reference.

According to the referee’s comment, a quotation has been included in the revised version of the manuscript.

Reviewer 1’s comment 12: Figure 15: Images provided are blurred. Please replace with higher quality images. Also, the sampling locations should be highlighted on the wooden panels.

According to the referee’s comment, the images have been replaced.

Reviewer 1’s comment 13: Section 4.3. For the FTIR analysis, please specify the spectral range as well.

According to the referee’s comment, spectral range has been included in the revised version of the manuscript in this section.

Reviewer 1’s comment 14: Lines 884-886. Please specify in what software was the FSD- deconvolution carried out.

According to the referee’s comment, software for FDS-deconvolution has been included in the revised version of the manuscript in this paragraph.

Reviewer 2 Report

Dear Authors,

I’ve read your wonderful work with a great interest. This is a case where culture and history “meets” high-tech chemical and mineralogical analyses of materials. Such multidisciplinary studies mark significant advance in our understanding of our heritage in board context. Your manuscript is based on in-depth research project, the outcomes of which are reported very appropriately. Methodological details and interpretations are very clear. The conclusions will be interesting to the broad circle of specialists. I strongly recommend this work for publication and specify below several recommendations for small amendments.

1)      Title: the expression “Analytical study” can be omitted. May be to add two-three words stating the general importance/novelty of this study?

2)      Abstract: please, shorten it a bit.

3)      Introduction: I think the first paragraph needs citations to some general historical sources (books or articles).

4)      Introduction: the last paragraph should start with concise statement of your objective.

5)      Subsection 3.1: is it possible to hypothesize about the possible sources of azurite?

6)      Subsection 3.2 (and/or below): I wonder whether climate changes (in regard to average temperature and moisture) have affected the studied object during several centuries after it was created.

7)      Discussion: can you add a new subsection speculating about possible conservation (even restoration) procedures and general management of this kind of heritage?

8)      Section 4: as this study mentions some minerals, I think it is reasonable to state that the present nomenclature of minerals adopted by IMA is followed. The actual formulae of minerals can be indicated in the text. See here for reference: http://cnmnc.units.it/ (select IMA list of minerals in the left navigation bar).

9)      Please, check the text again for correcting possible typos. For instance, Line 736: casar -> casal?

10)  Can you provide photos of entire casar and alfarje?

Author Response

Valencia, 19th January September 2023

Manuscript Number: 2160582

Title: Analytical study of a medieval polychromed alfarje found in a Hispano-Muslim casal (Palma de Mallorca, Spain)

Molecules

Corresponding author: Dr Carla Álvarez-Romero and Prof María-Teresa Doménech-Carbó

Dear Editor

            Thanks for your interesting comments and suggestions. According to them, we have modified the manuscript as follows:

Comments in the review letter:

Reviewer 2

Thanks for your valuable comments and suggestions that undoubtedly lead to improving the manuscript.

Reviewer 2’s comment 1: Title: the expression “Analytical study” can be omitted. May be to add two-three words stating the general importance/novelty of this study?

According to the referee’s comment, the tittle of the manuscript has been changed to “New insights into the medieval Hispano-Muslim panel painting. The alfarje found in a Balearic casal (Spain)”

Reviewer 2’s comment 2: Abstract: please, shorten it a bit.

According to the referee’s comment, the abstract has been shortened below 200 words.

Reviewer 2’s comment 3: Introduction: I think the first paragraph needs citations to some general historical sources (books or articles).

According to the referee’s comment, more references have been included in the revised version of the manuscript in this paragraph.

Reviewer 2’s comment 4: Introduction: the last paragraph should start with concise statement of your objective.

      According to the referee’s comment, the last paragraph in the introduction section has been more concisely re-written with clarification of the objectives of the study.

Reviewer 2’s comment 5: Subsection 3.1: is it possible to hypothesize about the possible sources of azurite?

According to the referee’s comment, a paragraph to hypothesize about the possible sources of azurite has been added.

Reviewer 2’s comment 6: Subsection 3.2 (and/or below): I wonder whether climate changes (in regard to average temperature and moisture) have affected the studied object during several centuries after it was created.

      According to the referee’s comment, the text has been enlarged including a short discussion on the possible effects of the climate change on the alfarje.

Reviewer 2’s comment 7:  Discussion: can you add a new subsection speculating about possible conservation (even restoration) procedures and general management of this kind of heritage?

According to the referee’s comment, a new subsection speculating about the possible conservation/restoration procedures and general management of this kind of heritage has been added.

Reviewer 2’s comment 8: Section 4: as this study mentions some minerals, I think it is reasonable to state that the present nomenclature of minerals adopted by IMA is followed. The actual formulae of minerals can be indicated in the text. See here for reference: http://cnmnc.units.it/ (select IMA list of minerals in the left navigation bar).

According to the referee’s comment, the present nomenclature of minerals adopted by IMA is followed.

Reviewer 2’s comment 9: Please, check the text again for correcting possible typos. For instance, Line 736: casar -> casal?

According to the referee’s comment, the text has been checked for correcting possible typos.

Reviewer 2’s comment 10: Can you provide photos of entire casal and alfarje?

According to the referee’s comment, photos of the casal and the alfarje have been provided.